# Medicine and Pharmacy Students’ Role in Decreasing Substance Use Disorder Stigma: A Qualitative Study

**DOI:** 10.3390/ijerph21121542

**Published:** 2024-11-21

**Authors:** Alina Cernasev, Rachel Barenie, Hayleigh Hallam, Kenneth C. Hohmeier, Shandra Forrest

**Affiliations:** 1College of Pharmacy, University of Tennessee Health Science Center, 301 S. Perimeter Park Drive, Suite 220, Nashville, TN 37211, USA; acernase@uthsc.edu (A.C.); khohmeie@uthsc.edu (K.C.H.); 2College of Pharmacy, University of Tennessee Health Science Center, 881 Madison Ave., Memphis, TN 38163, USA; 3College of Pharmacy, University of Tennessee Health Science Center, 1924 Alcoa Highway, Knoxville, TN 37920, USA; hhallam@uthsc.edu; 4College of Social Work, University of Tennessee, Knoxville, TN 37996, USA; sforres6@utk.edu

**Keywords:** SUD, interprofessional team, qualitative study, US

## Abstract

Background: A strong body of research has established stigma as a barrier to care for patients with substance use disorders (SUDs), which can lead to poorer patient outcomes. Prior qualitative research on healthcare practitioners’ perceptions is limited. This study aimed to describe healthcare professional students’ perceived roles in decreasing SUD stigma. Methods: A qualitative design using focus groups (FGs) was employed. This study applied the stigma conceptualization approach by Link and Phelan to develop the FG guide, including labeling, stereotyping, separation, status loss, and discrimination. These components are linked to the construction of cognitive categories that lead to stereotyped beliefs. The FG participants were graduate-level healthcare students recruited via email from the University of Tennessee Health Science Center (UTHSC). The research team analyzed the transcripts using Braun and Clarke’s approach to identify emergent themes in the data. Dedoose^®^ Version 9.0.107, a qualitative data analysis software platform, was utilized to facilitate data manipulation and retrieval during the analysis. Steps were taken to ensure the reliability of the qualitative data using Lincoln and Guba’s criteria. Results: Among thirty-one pharmacy and medical student participants, three themes emerged from the data: (1) student recognition of stigma, (2) the role of healthcare professionals in harm reduction, and (3) calls to enhance advocacy efforts to improve patient outcomes. These themes collectively encompass key members of the healthcare team’s perceptions and solutions to SUD stigma. Conclusions: This research reveals the importance of expanding training opportunities to go beyond the SUD disease state, to other evidence-based approaches such as effective advocacy, harm reduction, and stigma, which impact the delivery of that care.

## 1. Introduction

Stigma is a well established barrier to care for patients with substance use disorders (SUDs). It manifests in various ways, primarily from structural and social origins, whose intersectionality both reinforces and sustains stigma [1]. Structural stigma develops from public policies and laws, and it can be a major motivator for organizational policies that directly impact this group. Poor perceptions at the national and state levels can further influence public stigma and self-stigma, to which patients with SUDs are incredibly vulnerable [2]. Such ideas have been considered in the construction of multiple theoretical frameworks and approaches to stigma surrounding SUDs, which are utilized in prevention and treatment strategies targeted at this population [3].

Prior research has demonstrated that stigma can lead to poorer health outcomes for patients with an SUD, for a variety of reasons [4]. One important reason is that stigma held by the healthcare professional can influence the care that person provides to their patient. Providers may view this population as non-compliant, challenging, and a waste of healthcare resources [5]. These beliefs lead to consequences such as withholding certain services, undertreatment, and even discharging patients early due to concern that they will not adhere to a treatment regimen [6]. Research has found that these attitudes leave patients feeling like their medical needs are not sufficiently addressed, particularly mental health and social needs that influence outcomes after discharge [7].

Eliminating healthcare practitioner stigma surrounding SUDs is of paramount importance, and there are national, multidisciplinary efforts ongoing in the US working towards this goal [8,9,10]. General stigma reduction efforts in the past decade have primarily focused on interactions about and with those with SUDs. One daily practice includes removing terminology with negative connotations when discussing this group, including using “person-first” language [11]. Additionally, the use of personal narratives from this population helps to humanize these individuals, which may impact clinical decision-making [12]. However, in order to eliminate stigma through tailored and appropriate interventions, like those that are educational in nature, there must be an understanding of stigma related to SUDs among future healthcare professionals. This is because the next generation of healthcare professionals begin their training at an institute for higher education, and retaining trust in the healthcare system overall—especially the trust to manage one of the most stigmatizing disorders—begins as a student.

Prior research on this topic is limited but includes literature on healthcare students’ perspectives and perceptions of patients with SUDs and their related stigma, through several methods [13,14,15]. The results of those studies reveal that students across multiple disciplines are not confident in their ability to effectively interact with this unique population, particularly due to lack of understanding, experience, and training [13,14,15]. To quantify the extent of these feelings, one team in Japan developed a survey that measures the prevalence of stigma in healthcare professionals, serving as a reliable tool that can be replicated across various institutions [16]. Potential actions to address these results were captured by a recent systemic review on the stigmatizing attitudes of healthcare professionals and trainees, which found that most intervention studies saw a statistically significant decrease in stigma after study interventions [17]. More specifically, a study reviewing the impact of an SUD curriculum focused on reducing stigma found improved confidence among nurse practitioner students in caring for these patients [18].

These findings from the existing literature show promise in changing stigmatizing behaviors in the healthcare field, so long as mediation occurs early in career training. Based on these preliminary conclusions, we aimed to qualitatively evaluate healthcare professional students’ perceived roles in decreasing SUD stigma by conducting focus groups (FGs) of medical and pharmacy students. This practice allowed us to view perceptions of those with SUDs through a socio-cultural lens, which revealed opportunities for improvement in providing educational interventions at an earlier point in healthcare training.

## 2. Methods

### 2.1. Study Design

Qualitative methods—specifically, focus groups (FGs)—were used to achieve this objective instead of relying solely on traditional interviews. FGs can offer a deeper understanding of factors that influence health behaviors by fostering open discussions and helping participants feel more comfortable sharing their personal opinions [19]. This type of inquiry allows investigators to understand the socio-cultural factors surrounding stigma through the healthcare student lens [19,20]. This study utilized Link and Phelan’s stigma conceptualization approach to develop the FG guide [21]. Link and Phelan’s stigma conceptualization model consists of four components that guided the investigators not only in developing the FG guide but also in asking probing questions during the discussions [21]. These components include labeling, stereotyping, separation, and status loss and discrimination, and they are linked to the construction of cognitive categories that lead to stereotyped beliefs [21]. Link and Phelan recognize the stigma concept and its ability to affect multiple domains of an individual’s life in areas such as earning more income, obtaining housing, accessing healthcare, and other pillars of life itself [21]. To elicit change surrounding the vague definition of stigma, Link and Phelan address the need for a better understanding of the core issues of stigma research [21]. The idea of stigmatization is fueled by power, based on an individual’s access to social, economic, and political resources that permit the discovery of uniqueness among individuals, leading to unequal outcomes and access to opportunity [21]. A detailed description of our research methodology is presented elsewhere [13].

### 2.2. Recruitment

The participants were recruited via email from the University of Tennessee Health Science Center (UTHSC) for the FGs. The inclusion criteria for participants included being enrolled in a professional degree program at UTHSC at the time when the study was conducted, and being willing to participate in the study. Other eligibility criteria included speaking English and sharing their opinions about SUDs. The team continued recruiting participants and conducted focus groups until no new insights on the topic emerged from the discussions, achieving theoretical saturation [22].

At the beginning of the FGs, the researchers obtained informed consent from all participants. To safeguard confidentiality, the participants’ names were removed from the transcripts. The study protocol was approved by the Institutional Review Board (IRB) of the University of Tennessee Health Science Center (IRB: 21-07977-XM; 1 March 2021).

### 2.3. Data Analysis and Rigor

The research team analyzed the transcripts inductively using Braun and Clarke’s approach to identify emergent themes in the data [23]. The audio transcripts were sent for transcription to ensure an unbiased process. The verbatim transcripts were imported into Dedoose^®^ Version 9.0.107, a qualitative data analysis software platform, to facilitate data manipulation and retrieval during analysis. The research team independently read all of the transcripts to familiarize themselves with the data, and then two highly experienced coders coded all of the data inductively. The research team met to discuss conceptual differences regarding coding terminology and establish a common set of codes [24]. Each code received a description, and the team members reviewed and agreed upon the codes. The codes were grouped based on similarities into categories to identify key themes [23].

Steps were taken to ensure the reliability of the qualitative data using Lincoln and Guba’s criteria [25]. For example, a third party manually transcribed the de-identified FG data to avoid bias [25]. Prior to analysis, the research team read the corpus of the data and familiarized themselves [25]. The research team met regularly to evaluate the analyses critically and consider alternative views [25]. Thus, triangulation of the data was used to address validity concerns. Incorporating these steps improved the trustworthiness and credibility of the data, permitting the detection of variations in the data and the ability to resolve variations in interpretation [25].

## 3. Results

Five virtual focus groups were conducted, and a total of 31 participants attended. The average age was 27 years. Most of the participants were White (*n* = 19) and female (*n* = 21). Most of the participants were from the College of Medicine (COM) (*n* = 17), while the rest of the participants represented the College of Pharmacy (COP) (*n* = 14). Three themes emerged from the data, including (1) student recognition of stigma, (2) the role of healthcare professionals in harm reduction, and (3) calls to enhance interprofessional team activities to improve patient outcomes.

### 3.1. Theme 1: Student Recognition of Stigma

This theme revealed the participants’ perceptions of SUD stigma based on their background, upbringing, and implicit biases. Participants from the COM and COP touched on how stigma is shaped by societal attitudes and personal experiences, and how these perceptions may evolve over time. The FG discussions also touched on the complexity of treating substance abuse disorders, as they are often perceived as a choice rather than a disease, leading to challenges in providing appropriate care and support. Additionally, there was recognition of the role of societal teachings and media portrayals in perpetuating stigma, which can result in fear and reluctance to engage with patients seeking help. Overall, the participants emphasized the need for greater empathy, understanding, and de-stigmatization surrounding SUDs within both healthcare settings and society.

For example, during the FG discussion, one participant mentioned the perception of stigma:

“*chang[ing] depending on education level [and] depending on experience with populations with substance abuse…*”, as his perception of people with a SUD has evolved. He continued to say “*…[when] you come to college and then to medical school, and you get more highly educated… you see substance abuse disorders in a different light.*” (FG4, S1, COM) These FGs were successful in eliciting responses from students in healthcare about stigma and their perceptions of the opioid epidemic. While some students mentioned rural backgrounds framing their mindsets, others mentioned prejudice as the basis of stigma. The following excerpt echoes this sentiment: “*[belief] that stigma has a negative connotation because [she] always think[s] of stigma kind of synonymous with prejudice.*” (FG4, S2, COM) The implicit bias associated with SUD was further noted with the shared belief that SUD is regarded not as a disease, but rather as a choice with consequences. For one of the participants, he confirmed this experience through his upbringing in a “*rural small-town area of like 4000 to 5000 people, and the attitude was very much you’re doing this to yourself…*”, which leads to the outcast of this population from communities and worsening outcomes. (FG4, S1, COM). 

Again, SUD is a multifaceted disease state requiring approaches from physical, spiritual, medical, and emotional points of view to address the disease state and its progression. The participant shared his perspective of the disease state and approaching the care of these patients with “*compassion for people, and a need to understand that it’s a disease state that’s progressing.*” (FG4, S1, COM) The management of SUD is complex, since it does progress, so it is crucial to understand students’ perceived conceptions, since they “*carry throughout a group or a person that may be a part of that group*” and lead to generalizations. (FG2, S4, COM).

Generalizations may have resulted in fractured communities, complicating care for patients battling SUD. Another participant echoed the themes observed in small towns and shared instances where “*name calling*” occurred based on “*things that, if someone did use anything that wasn’t deemed as appropriate… and they were very much separated from the rest of society.*” *(*FG2, S5, COP).

Therefore, another participant stressed, it is important to “surround [your]self with people who can shape [your] stigma”, since “*…family and friends… can also have an impact on how [you] think about things as well.*” (FG4, S3, COP) In order to avoid stigma and “*look[ing] down on such a person because of the condition they have*” (FG1, S5, COP), like one participant mentioned, social teachings about the disorder should be increased. Another participant echoed the need for additional approaches “*because of the negativity related to the term and the media advertise it and always associate it with negative attitude*.” (FG1,S6, COP).

The evolution of negative connotations surrounding stigma is further seen in communities despite professional training among medical professionals. For example, the following excerpt discusses the “*…negative view on substance use disorders*” *and* “*the carry-over of negative* “*societal teachings… into their professional lives and their professional work.*” (FG1, S4, COM) When negative teachings alter the approach to standards of care, one participant emphasized instances when “*…patients and people who need assistance or need help get the bad end or the short end of the stick when it comes to substance use disorders and stigma.*” (FG1, S4, COM) Furthermore, another participant asserted that patients battling SUDs are “*associate[ed] with criminals*”, and this “*generates fear from the public*, *leading to further stigmatization of this population.*” *(*FG2, S6, COM).

### 3.2. Theme 2: The Role of Healthcare Professionals in Harm Reduction

This theme conveys participants’ understandings of the critical role played by healthcare professionals in harm reduction. Participants noted the utility of providing unused syringes to people who use drugs, so as to reduce the spread of communicable diseases such as HIV and hepatitis.

The following excerpt from one of the participants mentions being “*more open minded and liberal than some of the pharmacists that [she] works with…*” (FG2, S3, COP), which decreases the likelihood of a pharmacist with opposing views to sell syringes to a patient. This gap in care leads to the person “*inject[ing] whatever it is no matter what*” (FG2, S3, COP), and she would “*rather they have a clean needle and they be safe…*” (FG2, S3, COP) to reduce the spread of disease and mitigate public health threats.

There are needle exchange programs available to members of the public, where they can obtain clean syringes and supplies, as well as education about how to stay healthy. While programs exist, there is an overall lack of resources available for the public to access, so utilizing resources like auntbertha.com can be useful to link patients to the care they need. Another participant engaged in the FG discussion stated “*you can type in someone’s zip code and literally look at different types of programs and treatment programs…*” (FG2, S1, COP) and share the information with patients.

Barriers to care should be considered, but resources can address common themes seen among people battling SUDs and social determinants of health. Another barrier to care highlighted by a participant was the corporate regulation of the sale of syringes, as she commented on her corporate pharmacy’s stance on the issue with statements such as “*…you can’t sell this unless someone has proof of purchase… and so, if [the patient] had insulin, if [the patient] gets insulin with us, we have to call the doctor’s office and confirm they get insulin.*” (FG2, S1, COP).

Gaps in care like this lead to poor outcomes, and “*people are going to be doing it (injecting drugs) anyway…*”, so blocking or complicating access to clean supplies will likely lead to “*… a surge in HIV and hepatitis.*” (FG2, S1, COP).

### 3.3. Theme 3: Calls to Enhance Advocacy Efforts to Improve Patient Outcomes

This third theme highlights the necessity of enhancing approaches to addressing SUDs as an interprofessional team. Participants commented on various approaches to developing a successful interprofessional collaboration that improves patient outcomes. The participants highlighted that there is an urgent need for increased interprofessional interactions to address the care provided to patients with SUDs. Additionally, expanding access to care and resources such as unused syringes, supplies, and education is a crucial next step to address this epidemic.

For instance, one participant discussed her interactions with other members of the healthcare team and “*…[does not] know if [she] recall[s] and being successful because [she] feel[s] like , in a retail setting… all [we] can do are just like sow the seeds of getting help or [accessing] preventative measures, one being needle exchanges, and then Narcan…*” at disposal. She continued to advocate for resources and “*giving platforms and access to people who are those success stories and giving them the tools where they can go out and help people that were like them…*” (FG2, S1, COP). Again, there is a desperate need for increased access to care and resources to address the issue at hand. Another participant agreed that “*familiarity is a powerful tool*” and can encourage the growth of relationships between professionals and patients. When combined with compassion “*[familiarity] is just synergistic in the way that it can change somebody’s life*” and should be the foundation for a medical professional’s approach to providing care to this population. (FG2, S2, COM).

Another participant confirmed implicit bias as an emerging theme observed among healthcare providers in the USA, leading to patients “*…being stigmatized… [because] these patients go to a bunch of different pharmacies …*” to fill their pharmacotherapies for SUD. Many pharmacies “*…choose not to carry [substance abuse] medication because of the type of patients that it brings in*”, which widens the gaps in care seen among this population. (FG3, S6, COP) SUDs are time-consuming and complex to manage and understand; however, another participant asserted that “*…think[s] that just whenever you approach people, [he] always think[s] that, you know, coming from a non-adversarial or non-othering place is something that can get you on the right path to helping others…*” and can increase the level of compassion provided to the patient. (FG3, S5, COM).

It is easy to get lost in the role of a provider and the tasks of that role, and the participant encourages “*…seeing yourself in [the patients], as well as seeing themselves in you, [to] help build a strong relationship…*” with the patient that can lead to the beginning of their recovery. (FG3, S5, COM).

## 4. Discussion

Three themes emerged from this study: (1) student recognition of stigma, (2) the role of healthcare professionals in harm reduction, and (3) calls to enhance advocacy efforts to improve patient outcomes. These themes collectively encompass key members of the healthcare team’s perceptions and solutions to SUD stigma. This research presents an opportunity to train beyond the SUD disease state and on other evidence-based approaches, such as on effective advocacy, harm reduction, and stigma, which impact the delivery of that care.

A body of research has established that patients with SUDs are highly stigmatized, and healthcare practitioners’ perceptions can influence the type and quality of care that they receive. However, to date, limited data have described the perceptions about SUD stigma that students in health professions may hold. The findings from this study highlight that medical and pharmacy students come into their professional programs with little to no prior engagement with this population. Their knowledge is attributed to the environment the student grew up in, where it appears that rural environments may have cultivated greater stigmatizing behaviors and implicit bias towards patients with SUDs. Prior research has shown that physician bias against patients with opioid use disorders is greater in rural compared to urban areas [26]. Other research has highlighted specific barriers to rural communities in accessing care, such as lack of basic services, transportation, and provider and treatment program options [27]. As mortality related to drug misuse continues to increase in rural areas specifically, it is crucial to focus efforts for addressing stigma in programs where a majority of students come from this background [28]. One key method that could be explored is to introduce students to personal narratives of people with SUDs. It has been proven that humanizing the experiences of this patient population helps to better understand their experiences and reduce stigma [29]. This can be through various media, including hearing the speaker in person, reading written stories, or engaging in social media, highlighting an opportunity for healthcare educators [30,31]. Encouraging this exposure may help break down any previous student beliefs and stereotypes with regard to this population, as well as educating those who have no previous experience.

Another key aspect of comprehensive care for patients with SUDs is preventing harm that they could experience from other manifestations of the disease, such as loss of employment, development of other diseases, and more. Unfortunately, many of these patients do not receive proper support in these areas due to lack of awareness about where to go for services [2]. This is particularly important in rural areas, where formal substance use treatment and support programs, such as syringe service programs, are limited [27]. In order to mitigate this, healthcare practitioners must also be involved in evidence-based and easily accessible harm reduction efforts, such as distributing naloxone, providing unused syringes, and more. Current attitudes among practitioners towards these services vary, with the primary limitation being that equipping patients with known SUDs with these resources will further encourage substance use [32]. Optimistically, the findings from this study imply that many students seem to be aware of various resources to offer their patients, with an understanding of their benefits. Due to the vulnerability and hesitancy of patients who seek out harm reduction services, it is crucial to continue to develop positive attitudes towards these practices that are free of judgement and bias.

Finally, medical and pharmacy students appreciated the need to advocate for change to improve patient outcomes. Advocacy, whether for oneself or one’s patients, is key to improving patient care. Several aspects of SUD treatment are highly regulated at the national, state, and organizational levels, posing barriers to care [3]. The participants in this study highlighted examples of some gaps in legislation, including requirements for syringe dispensing. Possession of items, such as syringes, varies across state and company policies, with specific requirements needed to sell these items over the counter [32]. As a result, patients with SUDs have inconsistent access to this harm-reducing service. For example, one study found that 46% of North Carolina pharmacists reported store policies on non-prescription syringe sales, most frequently requiring patient proof of a medical necessity [33]. Additionally, despite the recent over-the-counter availability of Narcan, hesitancy still exists from patients and their social circles regarding seeking it out [34]. There is an opportunity for improved education to healthcare providers about the legal requirements and liability associated with certain practices to ensure that providers are empowered to practice to the full extent that the law allows. These are both areas for intervention that can be addressed by knowledge and education of current laws, as well as advocating for legislation that provides improved access to these resources for patients.

This study, for the first time, qualitatively describes healthcare students’ perspectives on harm reduction and what associated stigma and biases they might have. While the students that participated in this study appeared to have general knowledge about the issues surrounding patients with SUDs, they represent a small sample of future healthcare professionals currently training in an urban environment. Opportunities exist to further expand on SUD training and education at programs across the country to build confidence and competence in managing a vulnerable population. It is clear that addressing stigma towards adults with SUDs must be a multifaceted approach, from our healthcare students to community partnerships and political support.

### Limitations and Strengths

Some limitations must be noted when interpreting this study’s qualitative data. For example, this study’s qualitative nature may limit the generalization of the results to broader healthcare professional students’ views. Also, the average age of the participants was 27 years old and only students from two health professional schools (medicine and pharmacy) participated in this study. It is unknown whether younger students or students from other health professional schools have a different view of stigma. However, future research could significantly benefit from including other healthcare professional students to further understand a broader perspective that might facilitate improved patient outcomes. This could potentially improve our understanding of interprofessional partnerships in clinical environments and provide valuable insights into designing pedagogical activities that resemble clinical settings.

## 5. Conclusions

Among over thirty pharmacy and medical student participants, three themes emerged from the data, including (1) student recognition of stigma, (2) the role of healthcare professionals in harm reduction, and (3) calls to enhance advocacy efforts to improve patient outcomes. This is the first qualitative study to investigate the perceptions of students in doctoral-level health professional education related to SUD stigma. These themes collectively encompass vital members of the healthcare team’s perceptions and solutions to SUD stigma. This research presents an opportunity to train beyond the SUD disease state and on other evidence-based approaches, such as effective advocacy, harm reduction, and stigma, which impact the delivery of that care.

## Data Availability

Data is not available due to privacy and ethical restrictions.

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
