# Peer review of "Medicine and Pharmacy Students’ Role in Decreasing Substance Use Disorder Stigma: A Qualitative Study"

_ijerph, 2024, doi:10.3390/ijerph21121542_

Round 1

Reviewer 1 Report

Comments and Suggestions for Authors

In the introduction you need to develop a more theoretical framework for stigma, differentiate between different types of stigma and how these processes operate, you also need to give a little more about the different types of anti stigma interventions.  Build a more comprehensive arguement and illustrate understanding of the subject area. In the introduction give us a little more about the methodology and indication of the key finding. 

What was the epistemological positioning and the theoretical framework, qualitative work required an in depth description of the positioning of the team and what you were trying to achieve: please describe in detail and justify the use of focus groups: why were they chosen over interviews.

Identify what the initial themes were and illustrate some of research questions asked.  provide a table in the manuscript with high level themes and the supplementary codes under them.  

Findings, please separate out the findings from text were possible and could you give a little more about the gender and age of the participants please.  

In the discussion please start with the main finding and then give us the supplimentary ones, relate findings to current literature and identify what is known, confirmed or new about your findings.  Can you also discuss the implication of your findings for HE educations, practitioners and students. 

Comments on the Quality of English Language

I think the quality of english is fine

Reviewer 2 Report

Comments and Suggestions for Authors

Limited and small sample size needs further justification.

Typographical errors need to be corrected for precision.

Line 62 Prevalence

Line 68 Practitioner

Line 276 manifestation

Line 278 receive

Line 302 hesitancy

Comments on the Quality of English Language

Typographical errors, very few, are mentioned in the author's comments.

Reviewer 3 Report

Comments and Suggestions for Authors

This study investigates healthcare students' perceptions of their roles in reducing stigma associated with substance use disorders, a common barrier to effective patient care resulting in poorer outcomes for these patients.

Utilizing focus groups, the research identified three key themes: recognition of stigma, the healthcare professional's role in harm reduction, and the need for enhanced advocacy efforts. The findings underscore the necessity for expanded training that includes advocacy and stigma reduction, highlighting the potential for improving patient outcomes through a more informed and proactive healthcare workforce. This study contributes significantly to the field of health sciences by addressing a critical gap in understanding how future healthcare professionals can actively combat stigma, thereby fostering a more supportive environment for patients with SUD and enhancing the overall quality of care. The findings of this study are important to inform the training of professionals who deal with SUD as they are the ones who lead the way in advocating for the elimination of stigma as a barrier to care. As discussed this is especially true in rural areas since there is a lack of anonymity in these small communities. This aspect could be unpacked a bit more in this article.

The study is guided by a sound framework, namely, the Stigma Conceptualization framework which informs the data collection. It’s not clear though to what extent the framework supports the data analysis and interpretation. It’s also unclear what value the Dedoose software package added to the analysis.

The inclusion criteria of being prepared to share their opinion is an important one, but is it possible that it could have biased the sample since potential participants might have declined to participate knowing that their judgment of the patients would not be aligned with the purpose of the study. To this end, there is also no clear explanation for why only medicine and pharmacy students responded to the online appeal to participate. The researchers could investigate this further.

Comments on the Quality of English Language

Needs finer editing

Reviewer 4 Report

Comments and Suggestions for Authors

Dear authors,

Many thanks for submitting this paper and allowing me to review.

The paper is on a topic that is very relevant and currently very topical, not just in the US but wider and where ever stigma may present and for whatever conditions.

Overall the paper is well written and a number of valuable points are made.

A few comments:

The average age is reported as 27 (line 122) this seems to indicated more mature students. I would be interesting if younger students have a differing view and if stigma awareness training has been provided at a previous opportunity.

A further limitation in my opinion is the population of the focus groups, the individuals self selected to apply to join the focus groups which may bring in a bias as they may have had previous opinion or interest in the topic. The numbers were also fairly small with only 31 from 2 colleges again this could introduce a bias if there were 1-2 outliers as there would be misrepresenting the general student group.

The paper could benefit from a final grammar check partly for consistency. e.g. College of Medicine and Pharmacy (lines 123/124 are capitalised (and abbreviated) whereas on lines 130 and 131 they are not. There are also a number of "s" in the manuscript e.g. line 130, 153, 190, 193 etc. The plural is also used when referring to single participants e.g. line 183, 185. These could be tidied up.

The quotes are well used with key messages reported e.g. quote on lines 196 and 197 with a good message and positive response.

The suggestions on actions such as patients stories and comments on line 268 are a valuable suggestion as it is important that students realise everybody has a different story and history and they should not be considered the same.

I look forward to this manuscript being accepted for print and adding to the increasing evidence base.

Thank you

Comments on the Quality of English Language

Minor changes as suggested above in author comments
